# Clinical Outcomes and Return-to-Sport Rates following Fragment Fixation Using Hydroxyapatite/Poly-L-Lactate Acid Threaded Pins for Knee Osteochondritis Dissecans: A Case Series

**DOI:** 10.3390/biomimetics9040232

**Published:** 2024-04-13

**Authors:** Taichi Shimizu, Yoichi Murata, Hirotaka Nakashima, Haruki Nishimura, Hitoshi Suzuki, Makoto Kawasaki, Manabu Tsukamoto, Akinori Sakai, Soshi Uchida

**Affiliations:** 1Department of Orthopaedic Surgery, University of Occupational and Environmental Health, 1-1 Iseigaoka, Yahatanishi, Kitakyushu 807-0804, Fukuoka, Japan; taichishimizu@clnc.uoeh-u.ac.jp (T.S.); belltree@med.uoeh-u.ac.jp (H.S.); k-makoto@med.uoeh-u.ac.jp (M.K.); m-tsuka@med.uoeh-u.ac.jp (M.T.); a-sakai@med.uoeh-u.ac.jp (A.S.); 2Department of Orthopaedic Surgery, Wakamatsu Hospital for University of Occupational and Environmental Health, 1-17-1 Hamamachi, Wakamatsu, Kitakyushu 807-0024, Fukuoka, Japan; yoichi0928111@gmail.com (Y.M.); hirotakanakashima58@gmail.com (H.N.); haruking2468@gmail.com (H.N.)

**Keywords:** osteochondritis dissecans (OCD), knee arthroscopy, arthroscopic fragment fixation, hydroxy appetite poly-L-lactic acid (HA/PLLA)

## Abstract

Osteochondritis dissecans (OCD) of the knee is an uncommon injury in young active patients. There is currently a lack of knowledge regarding clinical outcomes and return-to-sport rates after fragment fixation surgery using hydroxy appetite poly-L-lactic acid (HA/PLLA) threaded pins for knee OCD among athletes. Our purpose was to investigate the clinical outcomes and return-to-sport rates following osteochondral fragment fixation using HA/PLLA pins for the treatment of knee OCD lesions among athletes. A total of 45 patients were retrospectively reviewed. In total, 31 patients were excluded, and 14 patients were included. Pre- and postoperative patient-reported outcome scores (PROSs), including the International Knee Documentation Committee (IKDC) score and Knee Injury and Osteoarthritis Outcome Scale (KOOS), were compared. In addition, patients were categorized into four groups according to postoperative sports status: higher, same, lower than preinjury, or unable to return to sports. The mean age was 14.4 years (SD 1.67). All patients were male. All PROSs significantly improved at 6, 12, and 24 months postsurgery compared to presurgery. 50% of the patients returned to sports at the same or higher level after surgery. Fragment fixation using HA/PLLA pins leads to favorable clinical outcome scores and high return-to-sport rates in the treatment of athletes with knee OCD.

## 1. Introduction

Osteochondritis dissecans (OCD) of the knee is an uncommon condition of subchondral bony separation that often occurs in young skeletally immature active patients [1]. However, the cause of this pathology remains unknown, and it is most often thought to be the result of repeated stress and trauma to the subchondral bone [2]. Subsequently, the bone under the cartilage becomes necrotic due to impaired blood flow, resulting in the separation and release of osteochondral fragments from the footprint as an unstable OCD (International Cartilage Research Society Grade 3 or 4) [3]. Since this fragment can sometimes become unstable, articular cartilage along the surface of the femoral condyle can rupture, leading to early osteoarthritis (OA) in young patients [4]. Currently, several therapeutic options, including refixation of the detached fragment, bone marrow stimulation (enhancing in situ healing), osteochondral autograft implantation, fresh osteochondral allograft transplantation, and cell-based or cell-free regenerative techniques, have been proposed for treating OCD lesions of the knee [4]. When fixing OCD fragments, non-absorbable screws were conventionally used as fixation materials [5,6]. However, bioabsorbable implants have been gaining increased traction as a favorable treatment option for OCD fragment fixation for more than 10 years [7]. Hydroxyapatite/poly-L-lactic acid (HA/PLLA) material has been previously used as a suture material to fill bone defects and to pin or plate bone fractures because it is absorbable and unnecessary to remove. To date, some case reports have described favorable clinical results after bioabsorbable implants were used to treat knee OCD [7,8,9,10].

We applied bioabsorbable pins made from HA/PLLA (Super-Fixsorb 30, Teijin Medical Technologies Co., Ltd., Osaka, Japan) for the fragment fixation of knee OCD lesions. A systematic review summarized the literature on OCD in children and adolescents fixed with absorbable pins, which are indicated for lesions that are unstable and narrow in extent [11]. It is made by mixing the bioabsorbable polymer PLLA with highly bioactive HA, which forms a void on the rough surface when observed in a cross-section under an electron microscope [12]. Forged reinforcements of HA/PLLA have been widely used in the orthopedic and dental fields as absorptive bone bonding agents [13]. However, it has been used primarily for the treatment of microfractures and osteochondral injuries because HA/PLLA has poor mechanical behavior under tensile stress [12,14]. In some preclinical studies, other substances have been added to increase the strength of HA/PLLA or to accelerate the degradation of complementary-reinforced composites [12,13]. A case–control study evaluated radiological findings after open reduction and internal fixation of knee OCDs using bioabsorbable pins and revealed that clinical failure occurred in 9.1% of all patients [15]. Additionally, Ishikawa et al. treated 13 patients with juvenile knee OCD arthroscopically with bioabsorbable pins, and three revision surgeries were performed because of the underestimation of lesion instability [16]. In these two case studies, Gunze’s material (diameter 1.5 mm, length 15 mm) was used. Overall, bioabsorbable pins were useful for the fixation of intra-articular osteochondral fragments, despite occasional failures due to difficulties in OCD treatment. We used a head pin similar to that used in our previous study of elbow OCD [17]. As shown in Figure 1, we used two HA/PLLA pins with diameters of 2 mm and 3 mm.

Previous studies have reported on the rate of return to sports after surgery for knee OCD, regardless of the surgical technique. In a study on osteochondral allograft transplantation for knee OCDs, Cotter et al. reported that 13 of 18 patients (72.2%) who underwent osteochondral allograft transplantation for the secondary treatment of knee OCD returned to sports at the same or higher level than before surgery [18]. In another study on OCD of the patellofemoral joint, Kramer et al. reported that 22 of 26 patients (85%) who underwent surgery for OCD returned to sports [8]. However, in addition to these two studies, there is currently a lack of knowledge regarding clinical outcomes and return-to-sport rates after fragment fixation surgery using HA/PLLA threaded pins, specifically regarding knee OCD athletes. The objective of this study was to investigate the clinical outcomes and return-to-sport rates following arthroscopic osteochondral fragment fixation using HA/PLLA pins for the treatment of knee OCD among athletes. It was hypothesized that the fixation of knee OCD using HA/PLLA threaded pins would provide favorable clinical outcomes and enable most athletes with knee OCD to return to sports at a higher level than before surgery.

## 2. Materials and Methods

### 2.1. Study Design and Setting

The present study is a retrospective case series. All procedures were performed at a single institute. This study was approved by the Ethics Committee of Medical Research (approval number UOEHCRB21-009) in accordance with the Declaration of Helsinki, 2013. Patients were identified via a retrospective query of the database based on the specific inclusion and exclusion criteria described in the next section. An associated imaging database was subsequently searched to identify the subset of patients with the necessary magnetic resonance images (MRIs). Demographic data, including age at the time of surgery, body mass index (BMI), sex, and preinjury sports participation, were collected. All patients had preoperative computed tomography (CT) scans and MRI. The size of each OCD lesion was measured by preoperative CT scans using multidimensional slices.

### 2.2. Participants

A total of 45 patients who underwent arthroscopic surgery to treat distal femoral OCD between October 2006 and March 2021 were retrospectively reviewed. Indications for surgery included positive findings for knee OCD lesions confirmed by physical examination; radiographic assessments, including X-ray, CT, and MRI; and failure of nonoperative treatments for more than 3 months. Exclusions were made for 26 patients presenting with International Cartilage Repair Society (ICRS) grade 4 bone fragments or those who had undergone additional surgical interventions like microfracture, osteochondral autologous transplantation, or autologous chondrocyte implantation. In addition, five patients were excluded due to loss of follow-up. As a result, 14 patients were enrolled in this study (Figure 2). Surgery was performed open or arthroscopically depending on the grade of cartilage damage, location, and size of the OCD lesion, as shown in Figure 3.

### 2.3. Surgical Technique

Prior to surgery, CT was used to assess the location of each OCD lesion. Each arthroscopic procedure was performed by a single surgeon (S.U). At the start of surgery, each patient was placed in a supine position under general or spinal anesthesia. Physical examination under anesthesia and arthroscopic evaluation of the meniscus, cartilage, and ligaments were performed. After an arthroscope was introduced into the knee joint, the instability of the OCD lesion was evaluated by probing. The free bony knee fragments were removed, and the subchondral bone in the damaged lesion was refreshed. If the bony fragments were enlarged, they were trimmed down to an adequate size with a Kirschner wire. If an OCD lesion required repair, a guide tube (DePuy Synthes Japan, Tokyo, Japan) was inserted through the surgical portals when viewing from a scope. A guide hole was then drilled to the appropriate depth using a custom-made dilator bit, and the drill guide was drilled to the desired depth using a Kirschner wire (Figure 5E). The drill hole was made a few millimeters deeper than the length of the HA/PLLA pin to minimize intraosseous pressure. Intermittent drilling was used to minimize temperature increases, which could cause thermal necrosis. The dilator was then inserted into the drill guide and tapped to the desired depth. Next, we inserted HA/PLLA pins (Super-Fixsorb 30, Teijin Medical Technologies Co., Ltd., Osaka, Japan) through the drill guide with a delivery tamp to fix the displaced or detached OCD lesion (Figure 5F). Two or three HA/PLLA pins were utilized to prevent rotation of the fixed fragment of the OCD lesion after surgery. Finally, the OCD lesion was probed to confirm rigid fixation.

### 2.4. Postoperative Rehabilitation

Range of motion (ROM) exercises were performed without restriction. In terms of weight-bearing, patients with OCD lesions within the non-weight-bearing range were allowed full weight bearing as tolerable starting from the day after surgery. Conversely, patients with lesions in weight-bearing zones commenced with one-third partial weight-bearing at two weeks post-operation and progressed to full weight-bearing by the fourth week.

### 2.5. Surgical Findings and Clinical Outcomes

Arthroscopic findings and surgical treatment data were obtained from each patient’s surgical records. Pre- and postoperative (6, 12, and 24 months) patient-reported outcome scores (PROSs), including the Knee Injury and Osteoarthritis Outcome Score (KOOS), International Knee Documentation Committee (IKDC) score, University of California Los Angeles (UCLA) activity score [19], and ROM measurements for the affected knee joint, were measured. Postoperative complications were retrospectively recorded.

### 2.6. Return to Sport

We conducted a thorough assessment of the patients’ ability to return to sport (RTS) after surgery, comparing it to their pre-surgery performance levels. We categorized the results into 4 groups (higher, same, lower than the presurgery level, or unable to return to sport) using a modified version of the return to preinjury sports activity level assessment established by the American Shoulder and Elbow Surgeons [20].

### 2.7. Postoperative Radiographs and MRIs

Postoperative knee radiographs were obtained monthly with anteroposterior, lateral, and skyline views and one CT scan up to 6 months after surgery. All patients underwent 3.0 T MRI at 12 months postsurgery to evaluate the quality of repaired cartilage healing using the magnetic resonance observation of cartilage repair tissue (MOCART) scoring system [21]. This system consists of 9 parameters: M1 (filling of the defect), M2 (cartilage interface), M3 (surface), M4 (OCD lesion structure), M5 (signal intensity), M6 (subchondral lamina), M7 (subchondral bone), M8 (adhesion), and M9 (effusion). Osseous union of the OCD lesions was confirmed by the absence of junctions between the fragments and host bone on plain radiographs, CT, and MRI scans [17].

### 2.8. Statistical Analysis

Dunnett’s multiple comparison test was utilized to evaluate the differences between preoperative PROS and the range of flexion against the scores obtained at 6, 12, and 24 months postsurgery. Wilcoxson’s signed rank test was applied to compare the preoperative and 12 months postoperative extension measurements. All statistical analyses were conducted using GraphPad Prism version 10.2.1, with a *p*-value threshold of less than 0.05 set for determining statistical significance.

## 3. Results

### 3.1. Patient Demographic Data and Surgical Records

Patient demographic data are shown in Table 1. The mean age was 14.4 ± 1.67 (12–17) years, and all 14 patients were male. The mean BMI was 19.6 ± 2.23 (15.9–24.2) kg/m^2^. The numbers of OCD lesions observed in the MFC (Medial Femoral Condyle), LFC (Lateral Femoral Condyle), and trochlear region were six, seven, and two, respectively.

Table 2 presents the surgical findings and types of procedures. The OCD lesion size was measured using preoperative CT scans. Two patients had an ICRS grade of 1, four patients had an ICRS grade of 2, and eight patients had an ICRS grade of 3. The mean follow-up after surgery was 16.7 ± 15.8 months (6–60). Four patients were assessed as skeletally immature because the epiphyseal plate of the distal femur was not closed (patients 1, 2, 7, and 8).

### 3.2. Clinical Outcomes

Preoperative and postoperative ROM measurements of the knees of patients with OCD were subsequently performed. The mean preoperative extension was −1.8 ± 5.4° greater than the postoperative extension. Although the mean knee extension ROM improved from −1.8 ± 5.4° to 0° at 12 months postoperatively, there was no significant difference (*p* < 0.05) between the pre- and postknee extension ROM results, as shown by Wilcoxon’s signed rank test. In contrast, the mean knee flexion angle improved significantly after surgery from 127.9 ± 23.7 (baseline preoperative) to 145.0 ± 3.9 (6 months postoperatively; *p* < 0.05), 147.1 ± 2.7 (12 months postoperatively; *p* < 0.01), 148.0 ± 2.7 (24 months postoperatively; *p* < 0.05), and 146.6 ± 4.2 (final follow-up postoperatively; *p* < 0.05).

Preoperative and postoperative PROSs are shown in Figure 4. Fourteen scores were compared using Dunnett’s multiple comparison test. The IKDC scores preoperatively and at 6, 12, and 24 months postoperatively and at the final follow-up were 48.7 ± 22.2, 91.2 ± 11.0, 84.4 ± 22.5, 91.5 ± 13.3, and 91.7 ± 11.2, respectively (preoperative versus 6 months postoperatively, *p* < 0.01; preoperative versus 12 months postoperatively, *p* = 0.06; preoperative versus 24 months postoperatively, *p* < 0.01; preoperative versus final follow-up, *p* < 0.01). The KOOS scores preoperatively and at 6, 12, and 24 months postoperatively and at the final follow-up were 61.8 ± 27.6, 95.8 ± 4.3, 93.9 ± 9.4, 98.8 ± 1.7, and 96.0 ± 3.4, respectively (the statistical analysis could not be performed because of missing data). The UCLA activity scores preoperatively and at 6, 12, and 24 months postoperatively and at the final follow-up were 4.3 ± 1.4, 8.9 ± 1.3, 8.6 ± 1.7, 9.4 ± 1.3, and 9.3 ± 1.2, respectively (preoperative versus 6 months after surgery, *p* < 0.01; preoperative versus 12 months after surgery, *p* < 0.01; preoperative versus 24 months postoperatively, *p* < 0.01; preoperative versus final follow-up, *p* < 0.01).

### 3.3. Return-to-Sports Rate

Table 3 shows the RTS after surgery and the time from surgery to RTS. In total, seven patients (50%) returned to sports at a higher level postsurgery, and seven patients (50%) returned to sports at the same level postsurgery compared to presurgery. The mean duration from surgery to RTS was 6.8 ± 2.1 months (range: 4–11 months).

### 3.4. Postoperative Imaging Analysis

Thirteen patients underwent MRI 1 year postsurgery. One patient did not receive a postoperative MRI scan because of a lack of convenience. The MOCART score and time to bone union for each patient are shown in Table 4.

The mean total MOCART score was 83.1 ± 16.9 (*n* = 13). In 6 of the 13 patients, the total MOCART score was 100. However, the total MOCART score was 55 for 2 of the 13 patients and 5 of the 13 patients in their 70s. Regarding the M1, M2, and M3 parameters, all patients achieved a full score (100). M4 macrophages (OCD lesion structure) exhibited homogenous results in 12 patients (92.3%) and heterogeneous results in 1 patient (7.7%). The M5 (signal intensity) T2* signal was normal in six patients (46.2%), nearly normal in five patients (38.5%), and abnormal in two patients (15.4%). The PD-TSE signal intensity was normal in six patients (46.2%), nearly normal in five patients (38.5%), and abnormal in two patients (15.4%). For M6 (subchondral lamina), the results were intact in 11 patients (84.6%) and not intact in 2 patients (15.4%). M7 (subchondral bone) exhibited intact results in 10 patients (76.9%) and nonintact results in 3 patients (23.1%). M8 showed evidence of adhesions; no patients were found to have adhesions. M9 indicated effusion; seven patients (53.8%) had effusion, and six patients (46.2%) had no effusion.

The mean time from surgery to union was 4.6 ± 1.4 months (3–7, n = 13). One patient was excluded because of previous revision surgery of OATS for nonunion.

We evaluated the correlation between the MOCART score and PRO score, the MOCART score and time to bone union, and duration to return to sports using Pearson’s correlation coefficient test. The MOCART score was significantly correlated with IKDC 12 months postsurgery (*p* < 0.05) and duration to return to sports (*p* < 0.05).

### 3.5. Case Presentation

We show two typical cases of preoperative images, surgical findings, and postoperative images (Figure 5 shows patient 10, and Figure 6 shows patient 4).

## 4. Discussion

Our findings demonstrated favorable clinical outcomes and return-to-sport rates after treating OCD knee lesions in athletes via the fragment fixation HA/PLLA pin surgical technique. This procedure provided significant improvement in the PROS and enabled all patients to RTS. Additionally, 61.5% of patients experienced RTS at a higher level than before surgery, as shown by the UCLA activity scores. The mean time from surgery to union was 4.6 ± 1.4 months (3–7, n = 13). The mean total MOCART score was 83.1 ± 16.9 (n = 13). A total MOCART score of 100 was achieved in 6 of the 13 patients. The total MOCART score was 55 for 2 of the 13 patients and in the 70s for 5 of the 13 patients.

Although this study showed positive clinical outcomes and RTS after fragment fixation with HA/PLLA pins, indications for this surgical technique must be carefully determined before surgery. Similar surgical indications are used to treat elbow OCDs. In a recent paper on elbow OCDs, Uchida et al. reported that HA/PLLA thread pin fixation was successful in patients with grade 2 to 4 elbow OCD [17]. In our study, HA/PLLA thread pin fixation was indicated for grade 1 to 3 knee OCD patients. This surgical technique may be applicable for ICRS grade 4 knee OCDs but should be carefully considered given that the knee is a load-bearing joint. According to the previous literature, if an ICRS grade 4 OCD lesion is concomitant with fresh bleeding, fragment fixation with HA/PLLA pins might be able to cure this pathology [22].

Several studies exist that show that various fragment fixation procedures originate from successful arthroscopic approaches for treating knee OCD lesions. In our study, we employed a composite material comprising HA dispersed within a PLLA matrix, as extensively described in the prior literature [23,24,25,26]. HA/PLLA composites have been extensively investigated in fundamental research, demonstrating the beginning of resorption within four weeks and complete degradation over a span of three years. The mechanical properties of our HA/PLLA composites, including their bending strength and modulus, have been characterized in the literature. Shikinami et al. reported a bending strength of approximately 270 MPa, which surpasses cortical bone along with a modulus of 12 GPa, comparable to cortical bone [23]. Notably, the impact strength of our utilized material, measuring at approximately 166 KJ/m^2^, is significantly higher than that of polycarbonate. The substance is composed of HA particles dispersed in a PLLA (Mv: 400 kDa) matrix [23]. Kocher et al. demonstrated the effectiveness of various fixation methods, including variable pitch screws and bioabsorbable tacks in the treatment of juvenile knee OCD [27]. They found comparable healing outcomes across different fixation techniques, suggesting the viability of our HA/PLLA threaded as an alternative to traditional fixation methods such as autologous bone pegs. Webb et al. reported favorable clinical outcomes after fragment fixation using bioabsorbable or metal materials, showing that osseous integration was confirmed in 15 of 20 knees (75%) [28]. Additionally, Millington et al. reported excellent clinical outcomes after fragment fixation using bioabsorbable implants, reporting that fragment union was confirmed in 12 of 18 (67%) unstable OCD lesions in knee joints, with six knees (33%) requiring subsequent surgery to remove a loose body and reconstruct the articular surface by osteochondral transplantation. They also reported a complication rate of 18% regarding the back-out and breakage of PLLA nails [29]. In another study on the treatment of juvenile knee OCDs, Tabaddor et al. reported the outcomes of the fragment fixation technique using bioabsorbable smart nails (made of hydroxyapatite and shaped like a metal nail), and 22 patients out of 24 (91.7%) had good results, such as improved PROS and closed physes [10]. Among our patients, 1 of 13 patients (7.7%) experienced nonunion because of breakage of the HA/PLLA pin and subsequently underwent osteochondral autograft transplantation. Hiramatsu et al. used absorbable pins to treat knee OCD and assessed the radiographic outcomes after surgery [15]. They reported a failure rate of 9.1% in their case–control study. Our surgical revision rate of 7.7% was similar to that in this study. Contraindications for this procedure included abnormally large lesions, OCD ICRS grade 3 patients, cystic lesions, and degenerative OCD lesion surfaces [22].

As one of the options for fragment fixation surgery, autologous bone pegs (or sticks), harvested from the proximal part of the tibia, are often utilized. Autologous bone pegs have several advantages compared to other fixation materials: (1) no need to perform secondary surgery to remove hardware; (2) little concern for foreign body reactions; and (3) a biologically enhanced healing process by autologous bone [30]. Although there is no information regarding RTS or the use of autologous bone pegs, previous studies have reported favorable outcomes of bone peg fixation for patients with knee OCD. However, a study by Slough et al. reported that of ten knee OCD patients treated with autologous bone peg fixation, four had partial defect healing, one had a tibial donor graft-site fracture, and 1 had their bone peg loosened during surgery [31]. Additionally, four patients required revision arthroscopy due to persistent knee pain. A total of four loose bodies, one loose peg, one meniscal tear, and one symptomatic hypertrophic synovium were observed among the patients. Compared to their study, our study showed fewer complications (2 out of 13 patients) and a lower revision rate (1 out of 13 patients). In addition, compared with autologous bone pegs, HA/PLLA pins do not require a harvesting process. The HA/PLLA pin fixation method is minimally invasive and has minimal donor site complications [32].

To date, there are no previous studies on RTS after fragment fixation for knee OCD lesions. In terms of RTS after knee OCD surgery, Cotter et al. reported RTS rates after osteochondral allograft transplantation for the secondary treatment of knee OCD [18]. Their results showed that 13 of 18 patients (72.2%) returned to sports at the same level or a level higher than that before surgery. In another study on RTS in OCD patellofemoral joint patients, Kramer et al. reported that 22 of 26 patients (85%) returned to sports postsurgery [8]. In our study, 100% of patients returned to sports at the same or higher level as before surgery. Our findings indicate that the HA/PLLA pin fixation technique might be more beneficial than other surgical procedures for treating knee OCD.

A previous systematic review of knee cartilage repair reported a correlation between each parameter of the MOCART score and clinical outcome measures [33]. They concluded that there was a correlation between autologous chondrocyte implantation (ACI) postoperative scores and MOCART scores. In our study, we evaluated the correlations between MOCART scores and PROSs (IKDC and KOOS), as well as between MOCART scores and the time to return to sport and the duration of bone union, using Pearson’s correlation coefficient test. We found a positive correlation between the MOCART score and the IKDC score at 12 months postsurgery and a negative correlation between the MOCART score and the time to return to the competition. However, the correlation between other PROSs and MOCART score has remained unclear. According to these results, MRI is a useful tool that can predict clinical outcomes and return-to-sport rates after knee OCD surgery.

This study has several limitations. First, the number of patients enrolled in this study was small because it was a single-center study, and the postoperative follow-up of some patients was short. Additionally, the minimum follow-up time for some patients was relatively short. Second, this was a case series and not a randomized controlled trial. Therefore, further evaluation of this surgical method with a larger cohort and long-term follow-up is needed. Third, the measurements were in agreement with those of the supervisor and the first author. Reliability was not calculated because it is not a continuous variable. Fourth, our study excluded patients who underwent surgical procedures other than bioabsorbable pinning, such as microfracture and osteochondral autologous transplantation, because each surgical procedure required a different surgical indication. Although we did not compare the postoperative outcomes among those with knee OCD who underwent different surgical procedures, the majority of patients had favorable postoperative outcomes and RTS rates. In this case series, the patients were only adolescents, and it is not clear what would happen if HA/PLLA fixation was performed on an older patient with OCD. The ROCK study group underwent a large multicenter cohort study of 1004 patients with knee OCD, and the patients were classified according to the ROCK classification. This classification is favorable for obtaining a detailed description of mobile or immobile lesions [34]. If we used their system, there would be additional information about arthroscopic findings for knee OCDs. The ROCK classification should be used in our future research on knee OCDs. In addition, the correlation between MOCART and IKDC and KOOS was unclear because of the limited number of included patients.

## 5. Conclusions

The arthroscopic fragment fixation surgical technique for treating knee OCD with absorbable hydroxyapatite/poly-L-lactic acid (HA/PLLA) threaded pins yields good clinical outcomes and favorable RTS rates in athletes. All patients returned to sports at a high level and were satisfied with their postoperative condition. The HA/PLLA pin surgical method is less invasive than the other methods and is a highly favorable option for athletes with knee OCD.

## Figures and Tables

**Figure 1 biomimetics-09-00232-f001:**
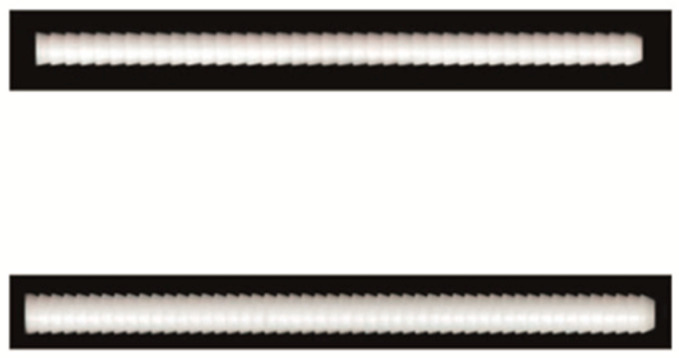
Two bioabsorbable pins with diameters of 2 mm (**top**) and 3 mm (**bottom**) were used. The pins, forged with 30% HA and 70% PLLA, are bioactive and bioabsorbable.

**Figure 2 biomimetics-09-00232-f002:**
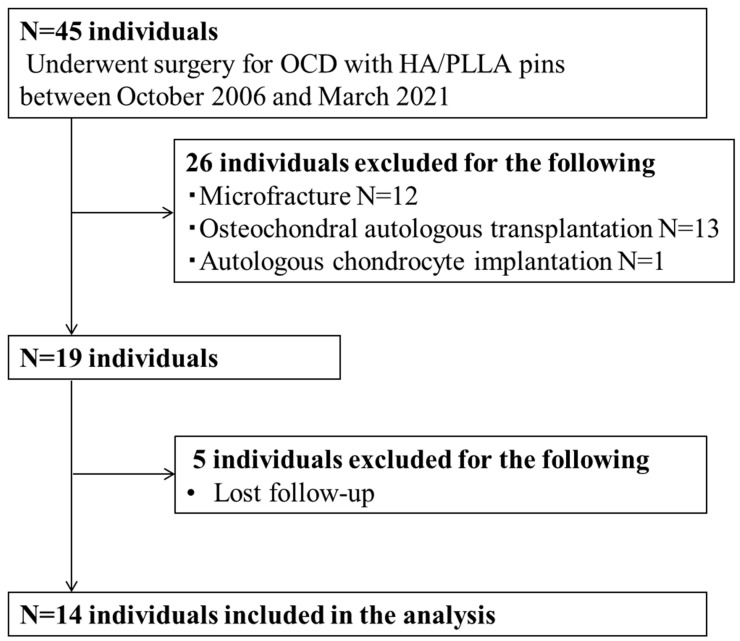
Flowchart showing the recruitment process for participants with knee OCD in this study. OCD: osteochondritis dissecans, HA/PLLA: hydroxyapatite/poly-L-lactic acid.

**Figure 3 biomimetics-09-00232-f003:**
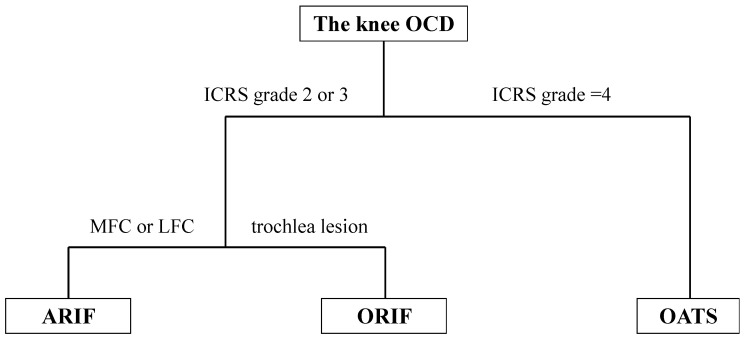
Classification of the ICRS grade according to the associated surgical procedure OCD: osteochondritis dissecans, ICRS: International Cartilage Repair Society, ARIF: arthroscopic reduction and internal fixation, ORIF: open reduction and internal fixation, OATS: osteochondral autograft transportation system.

**Figure 4 biomimetics-09-00232-f004:**
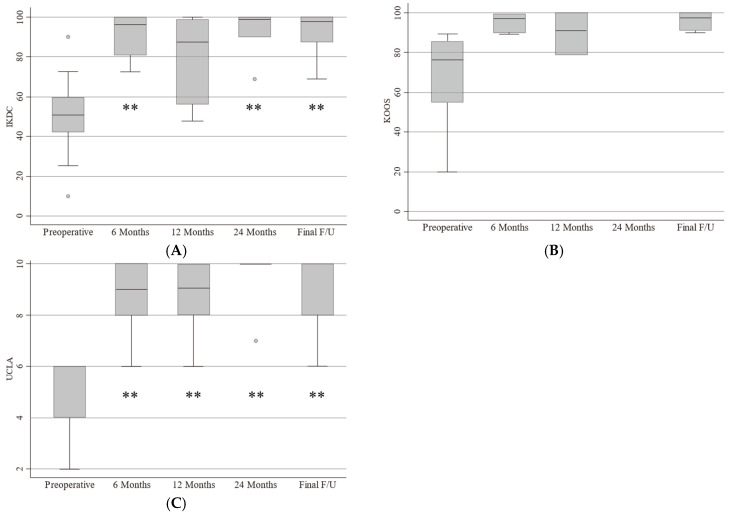
(**A**) Preoperative and postoperative PROSs. Pre- and postoperative IKDC scores; (**B**) KOOS scores; (**C**) UCLA activity score. **: *p* < 0.01. IKDC: International Knee Documentation Committee, KOOS: Knee Injury and Osteoarthritis Outcome Score, UCLA: University of California Los Angeles.

**Figure 5 biomimetics-09-00232-f005:**
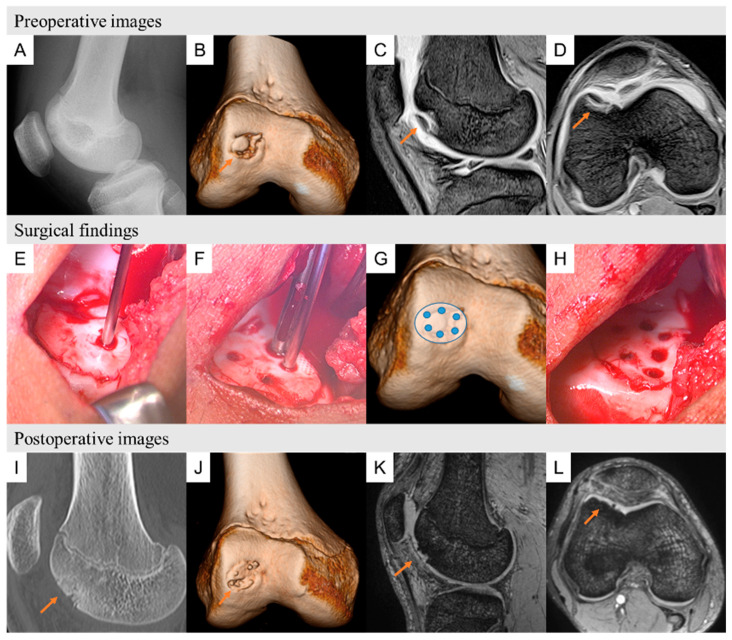
Patient 10 was a 16-year-old volleyball player suffering with right knee pain for the past year. (**A**) Plain lateral radiograph showing a round osteolytic lesion at the anterior aspect of the femoral condyle. (**B**) A three-dimensional computed tomography image showing an osteochondral lesion on the lateral side of the trochlea. (**C**) T2-star sagittal view showing cystic changes on the lateral side of the trochlea. (**D**) T2-star MR image showing cystic changes on the lateral side of the trochlea. Surgical findings of OCD repair using HA/PLLA pins. (**E**) Lateral longitudinal incision approaches. The OCD lesion was observed and then classified as ICRS grade 3 and temporarily fixed by a Kirschner wire. (**F**,**G**) CT images showing the locations of HA/PLLA threaded pins. (**H**) Fixed bone fragments with HA/PLLA pins of the same diameter. Postoperative images at 6 and 12 months postsurgery. (**I**,**J**) Sagittal view of computed tomography and three-dimensional computed tomography images at 6 months postsurgery showing complete bone union of the OCD lesions. (**K**,**L**) Magnetic resonance T2-weighted sagittal axial views at 12 months postsurgery showing neither effusion nor edema on the lateral side of the trochlea. OCD: osteochondritis dissecans, HA/PLLA: hydroxyapatite/poly-L-lactic acid, K-wire: Kirschner wire, OCD: osteochondritis dissecans, ICRS: International Cartilage Research Society, HA/PLLA: hydroxyapatite/poly-L-lactic acid.

**Figure 6 biomimetics-09-00232-f006:**
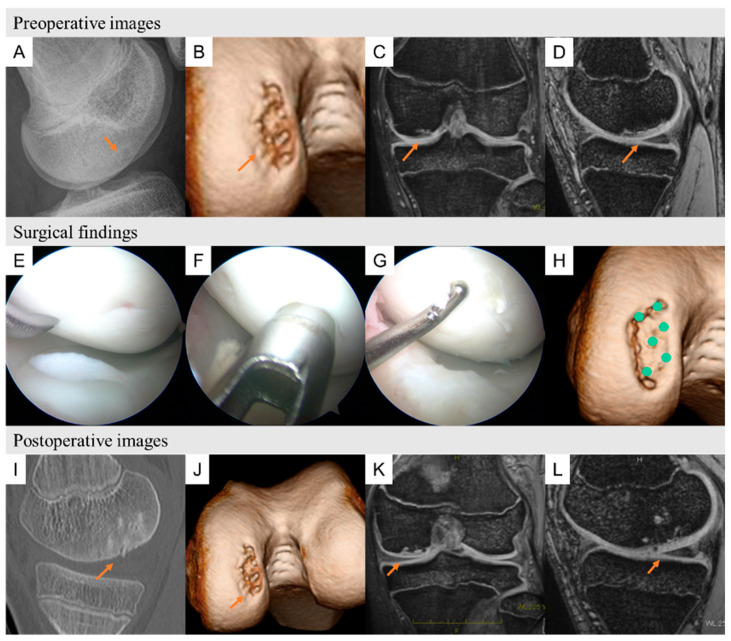
Patient 4 was a 14-year-old baseball player suffering with right knee pain for the past year. (**A**) Plain lateral radiograph showing a round osteolytic lesion at the anterior aspect of the femoral condyle. (**B**) A three-dimensional computed tomography image showing an osteochondral lesion on the lateral side of the trochlea. (**C**) T2-star sagittal view showing cystic changes on the lateral side of the trochlea. (**D**) T2-star MR image showing cystic changes on the lateral side of the trochlea. Surgical findings of OCD repair using HA/PLLA pins. (**E**) The OCD lesion was observed and classified as ICRS grade 2, and microfracture was performed along the edge of the lesion. (**F**,**G**) the bone fragments fixed with HA/PLLA pins. (**H**) CT image showing the HA/PLLA threaded pin locations. Postoperative images at 6 and 12 months postsurgery. (**I**,**J**) Sagittal view of computed tomography and three-dimensional computed tomography images at 6 months postsurgery showing complete bone union of the OCD lesion. (**K**,**L**) Magnetic resonance T2-weighted sagittal and axial views at 12 months postsurgery showing neither effusion nor edema on the lateral side of the trochlea. OCD: osteochondritis dissecans, HA/PLLA: hydroxyapatite/poly-L-lactic acid, K-wire: Kirschner wire, ICRS: International Cartilage Research Society.

**Table 1 biomimetics-09-00232-t001:** Patient demographic data. BMI: body mass index, ICRS: International Cartilage Repair Society, MFC: medial femoral condyle, LFC: lateral femoral condyle, OCD: osteochondritis dissecans.

Number	Age	Sex	BMI (kg/m^2^)	Sport	OCD Legion
1	14	male	24.2	baseball	MFC
2	12	male	18.7	soccer	MFC
3	16	male	19.5	basketball	MFC
4	16	male	17.1	basketball	MFC
5	13	male	18.6	baseball	MFC
6	14	male	19.2	basketball	MFC
7	12	male	15.9	soccer	LFC
8	13	male	19.5	hand ball	LFC
9	16	male	21.5	soccer	LFC
10	16	male	20.3	volleyball	LFC
11	17	male	22.6	soccer	LFC
12	13	male	16.7	volleyball	LFC
13	13	male	18.9	basketball	trochlea
14	16	male	21.8	soccer	trochlea

**Table 2 biomimetics-09-00232-t002:** Surgical findings and specific procedures. OCD; Osteochondritis dissecans, ICRS; International Cartilage Research Society grade, ARIF; arthroscopic reduction and internal fixation; ORIF; open reduction and internal fixation.

Number	Size of OCD (mm)	ICRS Grade	Procedure	Number of Pins
1	17 × 9 × 2.7	1	ARIF	3
2	30 × 15 × 3.0	2	ARIF	6
3	18 × 15 × 4.6	2	ARIF	5
4	20 × 13 × 6.3	2	ARIF	6
5	22 × 16 × 4.3	3	ARIF	5
6	20 × 20 × 2.9	3	ORIF	9
7	17 × 10 × 3.2	1	ARIF	2
8	16 × 14 × 5.6	2	ARIF	5
9	30 × 25 × 8.3	3	ARIF	9
10	15 × 13 × 6.7	3	ORIF	6
11	28 × 21 × 4.3	3	ARIF	6
12	25 × 15 × 3.7	3	ORIF	7
13	29 × 28 × 4.2	3	ORIF	10
14	26 × 19 × 3.9	3	ORIF	9

**Table 3 biomimetics-09-00232-t003:** The patient returned to sport level postsurgery. Level of competition postsurgery, time from surgery to return to competition. RTS: return to sport.

Number	RTS	Duration from Surgery to RTS (Month)
1	higher	4
2	same	8
3	higher	6
4	same	6
5	same	9
6	same	6
7	higher	5
8	same	11
9	same	10
10	higher	7
11	higher	8
12	higher	5
13	same	5
14	higher	6

**Table 4 biomimetics-09-00232-t004:** MOCART score and time to bone union for each patient. MOCART; magnetic resonance observation of cartilage repair tissue, M: month, M1: filling of the defect, M2: cartilage interface, M3: surface, M4: OCD lesion structure, M5: signal intensity, M6: subchondral lamina, M7: subchondral bone, M8: evidence of adhesion, M9: effusion.

Number	Time to Union (Month)	MOCART Score	M1	M2	M3	M4	M5 1	M5 2	M6	M7	M8
1	4	-									
2	3	100	20	15	10	5	15	15	5	5	5
3	3	75	20	15	10	5	5	5	5	5	5
4	6	75	20	15	10	5	5	5	5	5	5
5	4	75	20	15	10	5	5	5	5	5	5
6	4	70	20	15	10	5	5	5	0	0	5
7	5	100	20	15	10	5	15	15	5	5	5
8	6	75	20	15	10	5	5	5	5	5	5
9	-	55	20	15	10	5	0	0	0	0	5
10	4	100	20	15	10	5	15	15	5	5	5
11	7	55	20	15	10	0	0	0	0	0	5
12	3	100	20	15	10	5	15	15	5	5	5
13	4	100	20	15	10	5	15	15	5	5	5
14	7	100	20	15	10	5	15	15	5	5	5

## Data Availability

The data used to support the findings of this study are included within the article or available from the corresponding author.

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
