# Peer review of "Clinical Outcomes and Return-to-Sport Rates following Fragment Fixation Using Hydroxyapatite/Poly-L-Lactate Acid Threaded Pins for Knee Osteochondritis Dissecans: A Case Series"

_biomimetics, 2024, doi:10.3390/biomimetics9040232_

Round 1

Reviewer 1 Report

Comments and Suggestions for Authors

The results presented in the manuscript are of interest to a wide range of researchers. The use of transplants is an important task of regenerative medicine. The main disadvantage of this article is the almost complete absence of a description of the graft used.

1. The authors are recommended to justify the choice and use of this particular design and with such a composite composition.

2. Give a brief description of the properties of the structure used.

3. Provide links to preclinical studies.

4. In the "Discussion" section, conduct a comparative analysis of the use of the graft under study with similar structures.

5. What results could be expected when using this graft in older patients?

Author Response

Responses to the Reviewers’ comments

Reviewer 1

  1. The authors are recommended to justify the choice and use of this particular design and with such a composite composition.

Response: Thank you for your comment. Based on your suggestion, we have added the following description of HA/PLLA with references.

A systematic review summarized the literature on OCD in children and adolescents fixed with absorbable pins, which are indicated for lesions that are unstable and narrow in extent. [Wiktor, 2022]. (Lines 53-56)

  1. Give a brief description of the properties of the structure used.

Response: I agree with your comment. we have added the following sentences to the “Introduction” section according to your comment.

It is made by mixing the bioabsorbable polymer PLLA with highly bioactive HA, which forms a void on the rough surface when observed in cross-section under an electron microscope. [Takayama, Journal of Materials Science 2009] Forged reinforcements of HA/PLLA have been widely used in the orthopedic and dental fields as absorptive bone bonding agents. [Morizane, Mater Sci Eng C Mater Biol Appl 2019] However, it has been used primarily for the treatment of microfractures and osteochondral injuries because HA/PLLA has poor mechanical behavior under tensile stress. [Golan, Polymers 2021] (Lines 56-61)

  1. Provide links to preclinical studies.

Response: Thank you for your suggestions. We have referenced two basic experiments and have added descriptions as follows.

In some preclinical studies, other substances have been added to increase the strength of HA/PLLA or to accelerate the degradation of complementary reinforced composites. [Morizane, Mater Sci Eng C Mater Biol Appl 2019][Takayama, Journal of Materials Science 2009] (Lines 61-64)

  1. In the "Discussion" section, conduct a comparative analysis of the use of the graft under study with similar structures.

Response: We have added the following sentence as a comparative analysis.

Hiramatsu et al. used absorbable pins to treat knee OCD and assessed the radiographic outcomes after surgery. They reported a failure rate of 9.1% in their case‒control study. Our surgical revision rate of 7.7% was similar to that in this study. [Hiramatsu, Am J Sports Med 2024]. (Line342-344)

  1. What results could be expected when using this graft in older patients?

Response: In this case series, the patients were adolescents and it is not clear about what would happen if HA/PLLA fixation were performed on an older patient with osteochondritis dissecans. We have added the limitation regarding this issue.

In this case series, the patients were only adolescents, and it is not clear what would happen if HA/PLLA fixation was performed on an older patient with OCD. (Lines 397-398)

Reviewer 2 Report

Comments and Suggestions for Authors

Nice paper.

Well written article, perfect presentation and methodology.

pictures are presented the diagnosis ant operative method ad the result as well.

Comments on the Quality of English Language

well written and quality of english is fine

Author Response

Thank you for your fruitful comments.

Reviewer 3 Report

Comments and Suggestions for Authors

overall interesting topic, well designed and written paper. some issues still need to be solved:

-line 167: your statisistical test might be wrong: Wiconox test is not adequate for multiple comparisons at different time points. please seek aid from a proficient statistician

-line 104: "additional procedures". are these procedures done at the time of first surgery or are considered second surgeries? If this is the case, are these considered failures. please explain

-line 157: who assessed the images? what is the reliability of your measurments?

-minimal follow up time if very short (6 months), please explain this

-what is the re-operation rate? all cases were successfull?

-please correct figure numbers (figure 4 in the text is really figure 5). these correlations dont seem really significant; place consider deleting this figure

-there are two "figure 1". please pay attention

Comments on the Quality of English Language

acceptable

Author Response

Reviewer 2

  1. -line 167: your statisistical test might be wrong: Wiconox test is not adequate for multiple comparisons at different time points. please seek aid from a proficient statistician.

Response: Thank you for your suggestions. Dunnett’s test was performed according to the instructions of the statistical software. We have revised the following sentences. Please review the text on lines 178-183 and 207-228.

Before: Preoperative and postoperative ROM measurements of the OCD knees were performed. The mean preoperative extension was -1.8±5.4° compared to the postsurgery extension. Although the mean knee extension range of motion improved from -1.8±5.4° to 0° at 12 months postsurgery, there was no significant difference (p=0.32) between the pre- and post-knee extension range of motion results, as shown by the Wilcoxon signed rank test. In contrast, the mean knee flexion angle improved significantly after surgery from 127.9±23.7 (baseline presurgery) to 145.0±3.9 (6 months postsurgery), 147.1±2.7 (12 months postsurgery), and 148.0±2.7 (24 months postsurgery). The P values included the following: preoperative versus 6 months postsurgery (p=0.002), preoperative versus 12 months postsurgery (p=0.02), preoperative versus 24 months postsurgery (p=0.06), and preoperative versus 24 months postsurgery (p=0.02).

Preoperative and postoperative PROSs are shown in Figure 4. Fourteen scores were compared using the Wilcoxon signed rank test. All PMRs significantly improved after surgery. The IKDC scores at preoperative and at 6, 12, and 24 months postsurgery were 48.7 ± 22.2, 91.2 ± 11.0, 84.4 ± 22.5, and 91.5 ± 13.3, respectively (preoperative versus 6 months postsurgery, p= 0.003; preoperative versus 12 months postsurgery, p= 0.07; pre-operative versus 24 months postsurgery, p=0.04). The preoperative KOOS scores at 6, 12 and 24 months postsurgery were 61.8 ± 27.6, 95.8 ± 4.3, 93.9 ± 9.4, and 98.8 ± 1.7, respec-tively (preoperative versus 6 months postsurgery, p=0.02; preoperative versus 12 months postsurgery, p=0.18). The UCLA activity scores preoperatively and at 6, 12 and 24 months postsurgery were 4.3 ± 1.4, 8.9 ± 1.3, 8.6 ± 1.7, and 9.4 ± 1.3, respectively (preoperative ver-sus 6 months postsurgery, p=0.003; preoperative versus 12 months postsurgery, p=0.03; preoperative versus 24 months postsurgery, p=0.04).

After: Preoperative and postoperative ROM measurements of the knees of patients with OCD were subsequently performed. The mean preoperative extension was -1.8±5.4° greater than the postoperative extension. Although the mean knee extension range of motion improved from -1.8±5.4° to 0° at 12 months postoperatively , there was no significant difference (p < 0.05) between the pre- and postknee extension range of motion results, as shown by the Wilcoxon signed rank test. In contrast, the mean knee flexion angle improved significantly after surgery from 127.9 ± 23.7 (baseline  preoperative) to 145.0 ± 3.9 (6 months postoperatively ; p < 0.05), 147.1 ± 2.7 (12 months  postoperatively ; p < 0.01), 148.0 ± 2.7 (24 months postoperatively ; p < 0.05) and 146.6 ± 4.2 (final follow-up  postoperatively ; p < 0.05).

Preoperative and postoperative PROSs are shown in Figure 4. Fourteen scores were compared using Dunnett’s multiple comparison test. The IKDC scores preoperatively and at 6, 12, and 24 months  postoperatively  and at the final follow-up were 48.7 ± 22.2, 91.2 ± 11.0, 84.4 ± 22.5, 91.5 ± 13.3 and 91.7 ± 11.2, respectively (preoperative versus 6 months postoperatively , p < 0.01; preoperative versus 12 months postoperatively , p = 0.06; preoperative versus 24 months postoperatively , p < 0.01; preoperative versus final follow-up, p < 0.01). The KOOS scores preoperatively and at 6, 12, and 24 months postoperatively and at the final follow-up were 61.8 ± 27.6, 95.8 ± 4.3, 93.9 ± 9.4, 98.8 ± 1.7 and 96.0 ± 3.4, respectively (the statistical analysis could not be performed because of missing data). The UCLA activity scores preoperatively and at 6, 12, and 24 months postoperatively and at the final follow-up were 4.3 ± 1.4, 8.9 ± 1.3, 8.6 ± 1.7, 9.4 ± 1.3 and 9.3 ± 1.2, respectively (preoperative versus 6 months after surgery, p < 0.01; preoperative versus 12 months after surgery, p < 0.01; preoperative versus 24 months  postoperatively , p < 0.01; preoperative versus final follow-up, p < 0.01). (Lines 210-233)

  1. -line 104: "additional procedures". are these procedures done at the time of first surgery or are considered second surgeries? If this is the case, are these considered failures. please explain

Response: The term "additional procedures" was not appropriate.

As shown in Figure 1, we excluded 26 patients undergoing other procedures. The text was rewritten as follows.

Before: Patients who had a bone fragment of International Cartilage Repair Society (ICRS) grade 4 and/or who underwent other surgical procedures, such as microfracture, osteochondral autologous transplantation, or autologous chondrocyte implantation, were excluded. Subsequently, 26 patients who underwent additional surgical procedures were excluded.

After: 26 patients were excluded if they had an International Cartilage Repair Society (ICRS) grade 4 bone fragment and/or underwent other surgical procedures, such as microfracture, osteochondral autologous transplantation, or autologous chondrocyte implantation. (Lines 112-115)

3 -line 157: who assessed the images? what is the reliability of your measurements?

Response: Measurements were in agreement with the supervisor and the first author. The reliability was not calculated because it was not a continuous variable. The following statement was added to the limitation.

The measurements were in agreement with those of the supervisor and the first author. Reliability was not calculated because it is not a continuous variable. (Lines 390-391)

4 -minimal follow up time if very short (6 months), please explain this

Response: The minimum follow-up time of 6 months was short. We have added this information at the section of  the limitations.

First, the number of patients enrolled in this study was small because it was a single-center study, and the postoperative follow-up of some patients was short. Additionally, the minimum follow-up time for some patients was relatively short. (Lines 385-388)

5 -what is the re-operation rate? all cases were successfull?

Response: The reoperation rate stood at 7.7%, which included a (Refer Lines 269-270)

6 -please correct figure numbers (figure 4 in the text is really figure 5). these correlations dont seem really significant; place consider deleting this figure

Response: Figure 5 did not seem very meaningful, and we have removed this figure.

  • -there are two "figure 1". please pay attention

Response: Thank you for your remark. We have corrected the label number.

Round 2

Reviewer 1 Report

Comments and Suggestions for Authors

The authors are recommended to substantiate in more detail the relationship between the properties of the material and the results obtained. Since there is practically no description of the properties of the material in the manuscript, it is difficult to make comparisons with similar materials presented in the literature.

Author Response

March 23, 2024

Responses to the Reviewers’ comments

Reviewer 1

  1. The authors are recommended to substantiate in more detail the relationship between the properties of the material and the results obtained. Since there is practically no description of the properties of the material in the manuscript, it is difficult to make comparisons with similar materials presented in the literature.

Response: I agree with your comment. we have added the following sentences to the “Discussion” section according to your comment.

In our study, we employed a composite material comprising HA dispersed within a PLLA matrix, as extensively described in prior literature. [Shikinami, 1999][Hasegawa, 2002][Yasunaga, 1999] [Fukunaga, 2024] HA/PLLA composites have been extensively investigated in fundamental research, demonstrating the beginning of resorption within four weeks and complete degradation over a span of three years.  The mechanical properties of our HA/PLLA composites, including their bending strength and modulus, have been characterized in the literature.  Shikinami et al reported a bending strength of approximately 270 MPa, which surpasses cortical bone along with a modulus of 12 GPa, comparable to cortical bone. Notably, the impact strength of our utilized material, measuring at approximately 166 KJ/m2 is significantly higher than that of polycarbonate.[Shikinami1999]  The substance is composed of HA particles dispersed in a PLLA (Mv: 400kDa) matrix[Shikinami,1999]. Kocher et al. demonstrated the effectiveness of various fixation methods, including variable pitch screws and bioabsorbable tacks in the treatment of juvenile knee OCD.  [Kocher2007]. They found comparable healing outcomes across different fixation techniques, suggesting the viability of our HA/PLLA threaded as an alternative to traditional fixation methods such as autologous bone pegs.

Out utili(Line 328-343)